# Magnitude of risky sexual practice and associated factors among big construction site daily laborers in Bahir Dar city, Amhara Region, Ethiopia

Kalkikan Worku Mitiku[1], Muluken Azage Yenesew[2], Getasew Mulat Bantie[3], Amare Alamirew Aynie [3] *

1 Department of Midwifery, GAMBY Medical and Business College, Bahir Dar, Ethiopia, 2 College of Medicine and Health Science, Bahir Dar University, Bahir Dar, Ethiopia, 3 Faculty of Community Health, ALKAN Health Science Business and Technology College, Bahir Dar, Ethiopia

⊚ These authors contributed equally to this work.
* amarea0412@gmail.com

## Abstract

### Background

Young person's susceptibility to sexually transmitted infection has been consistently linked to intractable work places. In Ethiopia, different behavior related interventions had carried out to raise awareness of risky sexual behaviors and their consequences. However, there is still limited information on risky sexual practices.

### Methods

A cross-sectional study was conducted among the big construction site daily laborers from April 1, 2019 to May 30, 2019. A pre-tested questionnaire was used for data collection, and data were entered into Epidata and transported to SPSS for analysis. Independent variables with p < 0.05 in the multivariate analysis were considered to have a statistically significant association with risky sexual practice.

### Result

Among 627 sample respondents, the magnitude of risky sexual practice was found to be 24.2%. Labor workers who had a history of an STIs (AOR = 4.29; 95% CI: 2.56, 7.19), those who enjoy in the nightclubs (AOR = 2.33; 95% CI: 1.34, 4.08), those who started sex by peer pressure (AOR = 3.42; 95% CI: 2.06, 5.68), substance users (AOR = 2.03; 95% CI: 1.08, 3.82), those who were unable to read and write (AOR = 3.65; 95% CI: 1.41, 9.67), living independently (AOR = 3.71; 95% CI: 1.78, 7.77) and living with relatives (AOR = 3.12; 95% CI: 1.06, 9.20) were statistically associated with risky sexual practice.

### Conclusion

The magnitude of risky sexual practice among big constriction daily laborers was high in Bahir Dar City likely to increase their vulnerability to HIV. The findings of this study show the

**Data Availability Statement:** All relevant data are within the manuscript and the authors can provide any data if requested.

**Funding:** The authors received no specific funding for this work. All financial expenses had been covered by authors.

**Competing interests:** The authors have declared that no competing interests exist.

**Abbreviations:** AIDS, Acquired Immune Deficiency Syndrome; CSW, Commercial Sex Workers; HIV, Human Immune Virus; SPSS, Statistical package for social science; STI, Sexually Transmitted Infection; WHO, World Health Organization.

need of targeted HIV prevention plan to give due attention for daily laborers who had a history of an STI, enjoy in the night clubs, peer pressure, substance users, educational status and living arrangement. The Amhara Education Bureau and the city education department have to design ways to deliver at least a high school education.

## Introduction

Risky sexual practice is any sexual act that might put individual's social, physical and psychological health at risk [1]. It encompasses sexual behaviors like having unprotected sexual intercourse, early sexual initiation and having multiple sexual partners [2]. Risky sexual practice is usually associated with other risky behaviors like miss use of the substance, poor school performance and violence behaviors [3]. The four main types of outcomes that can arise from risky sexual practices are: unwanted pregnancy, sexually transmitted infection (STI, including HIV), physical and psychological impairment [4].

HIV/AIDS has become one of the world's most serious health development challenges of the last three decades. In 2018/2019 37.9 million [32.7 million– 44.0 million] people were living with HIV/AIDS and 1.7 million [1.4 million– 2.3 million) new infections were estimated worldwide. At the Eastern and southern African Countries, 20.6 million [18.2 million–23.2 million] people were living with HIV/AIDS and 800 000 [620 000–1.0 million] new HIV infections were also registered [5, 6].

In Ethiopia, big construction enterprises that usually engage young people are increasing from time to time. The young people usually migrate from rural areas to urban areas, which make them exposed to the new environment and exposes them for the new environment and interact with different people with a different culture [7]. These population groups are exposed to unstable sexual partner and fragmented social networks [6].

A study conducted in the Metema district on sexual behavior among migrant laborers revealed that, 74% of respondents had sexual intercourse with commercial sex workers (CSW), 49% had practiced transactional sex, 69% had multiple sexual partners in the preceding six months and 49% had unprotected sexual intercourse with CSWs [8].

Because of limitations of comprehensive reproductive health knowledge, adolescents and young girls are exposed to different reproductive health problems [9]. This population groups are usually engaged in unprotected sex which in turn leads to unwanted pregnancy, induced abortion, and different sexually transmitted diseases including HIV/AIDS [10].

A mixed method study conducted in Central Texas suggested that poverty, limited educational status, social isolation, high mobility, and limited access to health care were factors that put daily laborers at high risk of HIV/STI transmission [11].

Other different studies also documented that sociodemographic characteristics; like age, marital status, education status, occupational status, income and behavioral characters like watching pornographic movies, drinking alcohol, using hashish and level of awareness about HIV/AIDS were factors for risky sexual behavior [12–14].

In spite of the implementation of different behavioral change interventions, particularly regarding risky sexual behaviors by the government and non-governmental organizations to curb the transmission of HIV/AIDS in Ethiopia, studies indicate that risky sexual behaviors and risky sexual practices are still considerably high [15].

In Amhara Region, particularly in Bahir Dar City, big construction enterprises and their construction projects are increasing from time to time. These enterprises are known to employ large numbers of day laborers. Most of the daily labor workers came from a rural community

and live in a closed neighborhood in and around the construction sites away from their spouse and families.

The major motive for conducting this study was to provide specific evidence regarding the sexual practice of big construction site daily laborers so as to assist in the targeted planning of HIV/AIDS prevention strategies.

Most studies in Ethiopia were mainly focusing on risky sexual behaviors of respondents (behaviors of individuals' that might lead to do risky sex). But until finalization of this study, investigators did not find documented findings on the prevalence of risky sexual practices (practical acts that might put individuals' health directly at risk) and its determining factors.

Though there are some studies conducted on risky sexual behaviors of labor workers of different enterprises [8, 11], most studies conducted in Bahir Dar City had focused primarily on high school, college and university students [10, 14]. Hence, except one study that has been documented before seven years [15], investigators of this study did not find other documented research conducted on big construction site daily laborers until the finalization of the study.

Understanding the level of Sexual Practice and its factors among big construction site daily laborers is critical to design targeted HIV/AIDS prevention strategies to curb the spread of HIV infection [16].

Therefore, the main aim of this study was to determine the magnitude and associated factors of risky sexual practices among big construction site daily laborers in Bahir Dar city, Amhara Region, Ethiopia.

## Methods

### Study design and area

A cross sectional study was conducted from April 1, 2019 up to May 30, 2019 in Bahir Dar city, the capital city of Amhara regional state. Bahir Dar is located 564 km from Addis Ababa, the capital city of Ethiopia. It is one of the cities in Ethiopia in which investment and construction are rapidly growing which employs many young people for daily labor. Because of this expansion many young people migrating from rural areas and other towns for labor work. According to unpublished data of the city Municipal Office, there were 645 registered new construction sites by 2017/18 in the city. Among these, 56 of them were labeled as "big construction sites" as per the criteria of the Municipal Office.

### Sample size determination

The sample size was determined using a single population proportion formula by considering; 95% confidence level, 5% margin of error, 44.9% the prevalence of risky sexual behavior among daily laborers [15].

Thus, the sample size was calculated as:

$$n = \frac{Z_{\alpha/2}^2 (p)(1-p)}{d^2}$$

Where: n = Sample size required, P = prevalence, d = margin of error, $(Z\alpha/2)^2$ = 95% confidence level

$$n = \frac{(1.96)^2 (0.449(1-0.449))}{(0.05)^2} = 380$$

Since the study has involved multistage sampling technique, the calculated sample size was multiplied by 1.5 to manage the design effect.

n = 380 X 1.5 = 570

After adding 10% none response rate the final sample size was **627.**

## Sampling procedure

For the purpose of sampling, each of the 56 big construction sites was considered as clusters. As a simple rule of thumb, when using clusters for sampling it is advisable to include as many clusters, as possible (up to 20%) to enhance the accuracy of the result [17]. Thus, 11 big construction sites from the 56 were selected to take samples. Lists of daily labor workers were taken from selected big construction sites. After proportional allocation to each site, systematic sampling technique was employed to select the respondents.

## Eligibility criteria

**Inclusion criteria.** All labor workers who were working in the selected big construction sites were included in the study.

**Exclusion criteria.** Labor workers who had the problem of hearing and those who were absent in their work place during data collection period were not included in the study.

## Operational definitions

**Big construction sites.** Big construction sites were any type of building construction sites which were engaged in building constructions that have heights of 18 meters and above; It includes constructions like: industrial buildings, market mall construction, hotel construction, government office construction [18].

**Risky sexual practice.** Married individuals who reported to have sex out of marital partner, or single, divorced or widowed individuals who had more than one sexual partner in the last 12 months, or inconsistent condom use with a non marital partner, or unmarried female respondents reported unwanted pregnancy were regarded to have practiced a risky sexual practice [19].

## Data collection instruments and procedure

An interview administered questionnaire was developed to collect data from respondents. The questionnaire was first prepared in English then translated into Amharic (local language) for the interview. Four trained nurses and one health officer were involved in data collection and supervision respectively. Each selected respondent was interviewed in face-to- face interview bases.

## Data quality control and assurance

Before the actual data collection, the interviewer administered questionnaire was pretested on 5% of respondents working in construction sites that were not selected for the study. Clarity, wordings, logical sequence and skip patterns of the questionnaire were corrected and some amendments were made. Training was provided for the data collectors and the supervisor for one day. The collected data were reviewed and checked every day for completeness and clarity.

## Data processing and analysis

Data was entered into the Epi data version .3 .5 .1 Software by defining legal values for each variable and setting skip patterns and the data was validated and exported to SPSS version 23 for analysis. Logistic regression was conducted to identify the strength of association between independent variables and Risky Sexual Practice. To control the possible effect of confounding

variables, all variables were checked at bivariate regression on p values of <0.2 cut point. Independent variables with p-value of < 0.2 on bivariate regression were passed into multivariate regression. The backward logistic regression was used in multivariate regression, and variables with a p-value of < 0.05 on multivariate analysis were considered to have a statistically significant association with Risky Sexual Practice.

### Ethics approval and consent of participants

Ethical approval was obtained from GAMBY Medical and Business College, Research and Publication Office with the reference number of GC-221/2011. The support letter was obtained from the Amhara National Health Bureau and Municipal Office. The purpose of the study was explained to the study participants and informed verbal consent was obtained in the Amharic language from each respondent. Confidentiality of information was maintained by omitting any personal identifier from the questionnaire and the participants were well informed and the data was collected voluntarily.

## Result

### Socio-demographic characteristics

A total of 627 respondents with a response rate of 100% had participated in the study. The majority of respondents were male 373 (59.5%). Three hundred fifty-seven (56.9%) of respondents were above the age of 24 years; while 197 (31.4%) were in- between the age of 21–24 years. Regarding marital status 209 (33.3%) respondents were married; while 214 (34.1%) were unmarried. Regarding ethnicity, 486 (77.5%) of respondents were Amhara. The origin of 402 (64.1%) respondents was from a rural community. Concerning educational level, 202 (32.1%) respondents were unable to read and write, while 83 (13.2%) had completed 9th grade and above (**Table 1**).

### Sexual practice and pregnancy related variables of big construction daily labors

Among study participants, 508 (81%) have had sexual experience in their lifetime. Among those who had sexual experience, about 309 (60.8%) had started sexual practice at less than 20 years of age. And among respondents who had sexual experience 141 (27.8%) had experience of STI.

Among married respondents, 40 (20.5%) had practiced sexual intercourse outside their marriage. Sixty five (12.8%) of female respondents had practiced transactional sex to get a job. Among female respondents, 34(15.9%) had a history of unwanted pregnancy, and among these 26(76.5%) had terminated via abortion. The majority of respondents who had more than one sexual partner, who did sex with CSW and those who did transactional sex were not using condom (**Table 2**).

### Behavioral characteristics of respondents

One hundred fifty eight (31.1%) of respondents had started sexual practice because of peer pressure. Among these, a considerable number of respondents 104 (20.5%) enjoy in the night clubs. Among respondents who started sex, 102 (20.1%) of them were used to chew chat, 132 (26.0%) of the were used to watch pornographic movies, 303 (59.6%) of them were currently drinking alcohol and 70 (13.8%) of them were the substance (hashish/Shisha) users (**Table 3**).

**Table 1. Socio-demographic characteristics of big construction site daily laborers.**

| Variable | Category | Frequency | Percent |
|---|---|---|---|
| Sex | Male | 373 | 59.5 |
| | Female | 254 | 40.5 |
| Age | 18–20 | 73 | 11.6 |
| | 21–24 | 197 | 31.4 |
| | >24 | 357 | 56.9 |
| Marital status | Married | 209 | 33.3 |
| | In relationship | 204 | 32.5 |
| | *Unmarried | 214 | 34.1 |
| Religion | Orthodox | 464 | 74.0 |
| | Muslim | 125 | 19.9 |
| | **Others | 38 | 6.1 |
| Ethnicity | Amhara | 486 | 77.5 |
| | Oromo | 116 | 18.5 |
| | ***Others | 25 | 4.0 |
| Monthly Income | >2000 | 531 | 84.7 |
| (in Ethiopian Birr) | ≤ 2000 | 96 | 15.3 |
| Residence | Urban | 225 | 35.9 |
| | Rural | 402 | 64.1 |
| Educational level | Unable to read and write | 202 | 32.1 |
| | Able to read and write | 182 | 29.0 |
| | Primary (1-8th) | 160 | 25.5 |
| | Secondary and above (grade 9th and above) | 83 | 13.2 |

*Unmarried: Single, divorced and widowed

**Others: catholic, protestant

***Others: guragie, tigrie.

## Factors associated with risky sexual practice

To assess the presence and strength of association between independent variables and the dependent variable, bivariate and multivariate regression analysis was performed.

Accordingly, sex, marital status, ever had an STI, enjoy in the night clubs, watching pornographic movies, drink alcohol, peer pressure, substance use, educational status and living arrangement were statistically associated at bivariate regression. After controlling confounding variables multivariate logistic regression analysis was done and the history of an STI, enjoy in the night clubs, peer pressure, substance use, educational status and living arrangement were statistically associated with risky sexual practice at $p < 0.05$.

Respondents who had a history of STI were about 4.30 times more likely (AOR = 4.29; 95% CI: 2.56, 7.19) to practice risky sex than their counterparts. Respondents who enjoy in the night clubs were 2.33 times more likely (AOR = 2.33; 95%CI: 1.34, 4.08) to practice risky sex than those who didn't enjoy the night clubs. In a similar manner, respondents who started sex by peer pressure were 3.42 times more likely (AOR = 3.42; 95% CI: 2.06, 5.68) to practice risky sex than their counterparts. Labor workers who were using substances were about two times more (AOR = 2.03; 95%CI: 1.08, 3.82) likely practices risky sex compared to the non users. Regarding educational status, respondents who were unable to read and write were 3.65 times (AOR = 3.65; 95% CI: 1.41, 9.67) more likely to practice risky sex compared to those who had grade 9th and more educational status. Respondents who were living independently and with their relatives were 3.71times (AOR = 3.71; 95% CI: 1.78, 7.77) and 3.12 times (AOR = 3.12;

**Table 2. Sexual practice and pregnancy related variables among study participants of big construction daily laborers.**

| Variables | Category | Frequency | Percent |
|---|---|---|---|
| Ever had sex | Yes | 508 | 81.0 |
| | No | 119 | 19.0 |
| Age at first sex | <20 | 309 | 60.8 |
| | ≥20 | 199 | 39.2 |
| Had a history of STI | Yes | 141 | 27.8 |
| | No | 367 | 72.2 |
| Times attend to religious | Daily in the evening and/or morning | 134 | 21.4 |
| | Once in week on Sunday | 330 | 52.6 |
| | Sometimes/ not on defined date | 163 | 26.0 |
| Married did sex outside marriage | Yes | 40 | 20.5 |
| | No | 155 | 79.5 |
| Condom use while having sex outside marriage | Yes | 13 | 32.5 |
| | No | 27 | 67.5 |
| Number of sexual partners in the last 12 months | One | 388 | 76.4 |
| | More than one | 120 | 23.6 |
| Use of condom while having sex with more than one partner | Yes | 90 | 75.0 |
| | No | 30 | 25.0 |
| Having sex with commercial sex workers | Yes | 72 | 14.2 |
| | No | 436 | 85.8 |
| Condom use while having sex with CSW | Yes | 42 | 58.3 |
| | No | 30 | 41.7 |
| Having transactional sex* | Yes | 65 | 12.8 |
| | No | 443 | 87.2 |
| Used condom while doing transactional sex | Yes | 37 | 56.9 |
| | No | 28 | 43.1 |
| Unwanted pregnancy | Yes | 34 | 15.9 |
| | No | 180 | 84.1 |
| Educational level of respondents who started sex | Unable to read and write | 165 | 32.5 |
| | Able to read and write | 146 | 28.7 |
| | 1-8th grade completed | 120 | 23.6 |
| | Greater than 9th grade | 77 | 15.2 |
| Unwanted pregnancy terminated in | Abortion | 26 | 76.5 |
| | Birth | 8 | 23.5 |
| Risky sexual practice | Yes | 123 | 24.2 |
| | No | 385 | 75.8 |

* Transactional sex = Doing sex to get jobs and/wage.

Among big construction enterprise day labors that had practiced risky sexual practice, the majority of them 67 (54.5%) were male (**Fig 1**).

95% CI: 1.06, 9.20) more likely to practice risky sex compared to those who were living with their parents (**Table 4**).

## Discussion

The result of this study revealed that the prevalence of risky sexual practice among daily laborers was 24.2%. The result shows that labor workers are highly vulnerable for risky sexual practice like long truck drivers, college and university students, and commercial sex workers [20, 21].

**Table 3. Behavioral characteristics of big construction site daily laborers, Bahir Dar in city.**

| Variable | Category | Frequency | Percent |
|---|---|---|---|
| Peer pressure to do practice | Yes | 158 | 31.1 |
| | No | 350 | 68.9 |
| Enjoy in the night clubs | Yes | 104 | 20.5 |
| | No | 404 | 79.5 |
| Currently chew chat | Yes | 102 | 20.1 |
| | No | 406 | 79.9 |
| Recently watching pornographic movies | Yes | 132 | 26.0 |
| | No | 376 | 74.0 |
| Those who started sex have awareness about HIV transmission | Yes | 485 | 95.5 |
| | No | 23 | 4.5 |
| Discuss with family about sexual issues | Yes | 173 | 34.1 |
| | No | 335 | 65.9 |
| Currently drink alcohol | Yes | 303 | 59.6 |
| | No | 205 | 40.4 |
| Currently use the substance (hashish and Shisha) | Yes | 70 | 13.8 |
| | No | 438 | 86.2 |
| Living arrangement | Independently | 306 | 59.8 |
| | With Parents | 102 | 21.5 |
| | With Boy/girl friends | 64 | 12.6 |
| | With Relatives | 36 | 6.1 |

The result was more or less consistent with other studies, like in Arba Minch, 22.4% [14] Asella town, 20.07% [22].

The prevalence of Risky Sexual Practice revealed by this study was less than a study conducted in Bahir Dar 44.9% [15], the disparity of the result might be due to the difference in focused concept of study. The previous study was conducted on Risky Sexual Behaviors, while the present study has concerned only with Risky Sexual Practices.

The result of the present study was also less than other study conducted in Bahir Dar Cit 40.6%, [23] and Mizan Amman 30.5% [24], but slightly higher than a study conducted in Aksum, 19.6% [25]. The disparity might be because of the difference in the focused population groups of the previous studies; a study in Bahir Dar City was focused on Private College Students; and studies in Mizan- Amman and Aksum were conducted on high school and

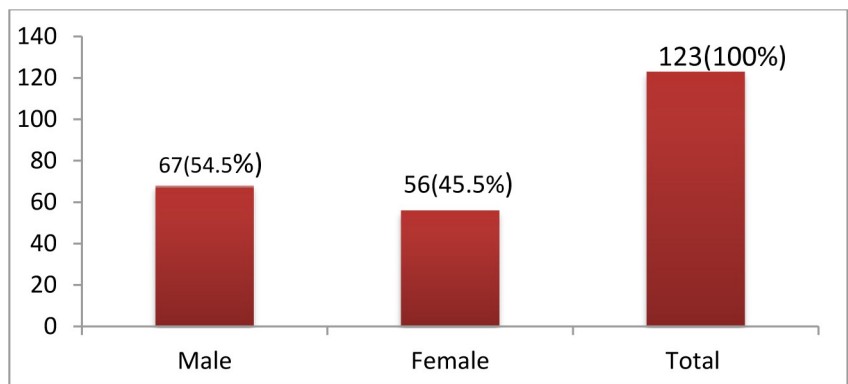

**Fig 1. Sex distribution of risky sexual practice among big construction site daily laborers in Bahir Dar in city.**

**Table 4. Factor analysis of predictor variables for risky sexual practice among big construction site daily laborers, Bahir Dar city.**

| Variable | Category | Risky Sexual Practice | | COR (95% CI) | AOR (95% CI) | P-value |
|---|---|---|---|---|---|---|
| | | Yes | No | | | |
| **Sex** | Male | 67 | 257 | 1 | 1 | |
| | Female | 56 | 128 | 1.68(.11, 2.54)* | 0.70(0.38,1.29) | |
| Marital status | Married | 41 | 165 | 1 | 1 | |
| | In relationship | 37 | 143 | 1.04(0.63, 1.71) | 0.70(0.37,1.33) | |
| | Unmarried | 45 | 77 | 2.35(1.42, 3.89)* | 1.11(0.65,2.26) | |
| History of STI | Yes | 70 | 76 | 5.37(3.47, 8.31) | 4.29(2.56,7.19)** | < 0.001 |
| | No | 53 | 309 | 1 | 1 | |
| Enjoy in the night clubs | Yes | 39 | 65 | 2.29(1.44, 3.64)* | 2.33(1.34,4.08)** | 0.003 |
| | No | 84 | 320 | 1 | 1 | |
| Watch pornographic movies | Yes | 46 | 86 | 2.08(1.34,3.22)* | 1.04(0.57,1.87) | |
| | No | 77 | 299 | 1 | 1 | |
| Ever drink alcohol | Yes | 89 | 214 | 2.09(1.34,3.26)* | 1.45(0.85,2.26) | |
| | No | 34 | 171 | 1 | 1 | |
| Peer pressure to practice sex | Yes | 72 | 86 | 4.91(3.19,7.56)* | 3.42(2.06,5.68)** | < 0.001 |
| | No | 51 | 299 | 1 | 1 | |
| Substance use | Yes | 26 | 44 | 2.08(1.22, 3.55)* | 2.03(1.08,3.82**) | 0.029 |
| | No | 97 | 341 | 1 | | |
| Educational status of respondents who started sex | Unable to read & write | 40 | 117 | 3.26(1.50, 7.05)* | 3.65(1.41,9.67)** | 0.009 |
| | Able to read & write | 39 | 109 | 2.83(1.29,6.19)* | 2.68(0.99,7.24) | |
| | 1-8th grade | 30 | 99 | 1.82(0.80,4.17) | 1.23(0.43,3.53) | |
| | Grade 9 & above | 14 | 60 | 1 | 1 | |
| Living arrangement | Independently | 86 | 220 | 2.93(1.53,5.63)* | 3.71(1.78,7.77)** | < 0.001 |
| | With Relatives | 12 | 24 | 3.75(1.50,9.40)* | 3.12(1.06,9.20)** | 0.039 |
| | With Boy/girl friends | 13 | 51 | 1.91(0.81,4.50) | 1.72(0.65,4.57) | |
| | With Parents | 12 | 90 | 1 | 1 | |

COR: Crude Odds ratio; AOR: Adjusted odds ratio

* significant at P < 0.2

** Significant at p < 0.05.

preparatory school students. Because of this, variables like the history of transactional sex was not included in the previous studies. The other reason would be due to the difference in the socioeconomic status of respondents and due to the difference in the living arrangement of respondents.

This study documented that factors like history of an STI, enjoying in the night clubs, and starting sex by peer pressure, educational status, and living style was associated with risky sexual practice.

This study showed that the male daily labor workers had practiced risky sex than female. Of 123 respondents who had practiced risky sexual practice, 67 (54.5%) were males; while 56 (45.5%) were females. Since the prevalence of HIV infection among women is higher than men [26, 27] most NGOs and governmental organizations might focus to empower female than male. Because of this male may be more reluctant than female.

The present study revealed that respondents who had a history of STI were 4.75 times more likely to practice risky sex compared to those who had no history of an STI. This result shows the presence of the strong relationship between sexual practice and STIs [28, 29].

In our study the participants who enjoy in the night clubs were 2.35 times more likely practice risky sex than those who did not visit night clubs. Alcohol and substance use is widely practiced in night clubs; and a majority of nightclubs employee young CSW to attract males. Therefore, visiting nightclubs exposes for the risky sexual practice [23, 30]

The present study also indicated that participants who started sex through peer pressure were 3.42 times more likely to practice risky sex than those who had started without peer pressure. This might be because most young people are pushed by other people who are with similar age groups than any other groups; and findings indicated that peer pressure is a factor for risky sexual practice [31].

Respondents at low grade (less than grade 5th) educational status were also more likely to practice risky sex compared to those whose educational status were greater than grade 9th. Awareness and educative massages about sexual issues from health professionals, NGO's and the government organizations is mostly via flyers or bulletin board posts, which need an educated community. But the majorities of labor workers in this study were unable to read and write; and hence might have a lower understanding of massages [32].

Regarding living arrangement respondents who were living independently were (3.71 times) and with their relatives (3.12 times) more likely to practice risky sex compared to respondents living with their parents. Daily labor workers who were living away from their parents begin their living in a new environment; and this leads to start a relation with new sex partners and this in turn leads do risky sexual practice [31].

## Conclusion

The magnitude of risky sexual practice among big constriction daily laborers was high. Ever had a history of an STI, enjoying in the night clubs, starting sex through peer pressure, educational status and living arrangement were statistically associated factors with risky sexual practice. The Bahir Dar city health department has to include these factors in its HIV prevention and health promotion activities to reduce risky sexual practices. The Amhara Education Bureau and the city education department have to design ways to deliver at least a high school education for young population.

## Limitation of the study

The study had included daily labor workers only from big construction enterprises. It doesn't include labor workers from any other business and small construction enterprises.

## Supporting information

**S1 File.**
(RAR)

## Acknowledgments

We would like to thank our data collectors and the supervisor for their invaluable effort; our deep gratitude also goes to our study participants who volunteered and took their time to give us all the relevant information for the study. Last but not least, we would like to thank all the big construction enterprise managers for their cooperation and support during the data collection.

## Author Contributions

**Conceptualization:** Kalkikan Worku Mitiku, Muluken Azage Yenesew, Getasew Mulat Bantie, Amare Alamirew Aynie.

**Data curation:** Kalkikan Worku Mitiku, Muluken Azage Yenesew, Getasew Mulat Bantie, Amare Alamirew Aynie.

**Formal analysis:** Kalkikan Worku Mitiku, Muluken Azage Yenesew, Getasew Mulat Bantie, Amare Alamirew Aynie.

**Investigation:** Kalkikan Worku Mitiku, Muluken Azage Yenesew, Getasew Mulat Bantie, Amare Alamirew Aynie.

**Methodology:** Kalkikan Worku Mitiku, Muluken Azage Yenesew, Getasew Mulat Bantie, Amare Alamirew Aynie.

**Project administration:** Kalkikan Worku Mitiku, Muluken Azage Yenesew, Getasew Mulat Bantie, Amare Alamirew Aynie.

**Resources:** Kalkikan Worku Mitiku, Muluken Azage Yenesew, Getasew Mulat Bantie, Amare Alamirew Aynie.

**Software:** Kalkikan Worku Mitiku, Muluken Azage Yenesew, Getasew Mulat Bantie, Amare Alamirew Aynie.

**Supervision:** Kalkikan Worku Mitiku, Muluken Azage Yenesew, Getasew Mulat Bantie, Amare Alamirew Aynie.

**Validation:** Kalkikan Worku Mitiku, Muluken Azage Yenesew, Getasew Mulat Bantie, Amare Alamirew Aynie.

**Visualization:** Kalkikan Worku Mitiku, Muluken Azage Yenesew, Getasew Mulat Bantie, Amare Alamirew Aynie.

**Writing – original draft:** Kalkikan Worku Mitiku, Muluken Azage Yenesew, Getasew Mulat Bantie, Amare Alamirew Aynie.

**Writing – review & editing:** Kalkikan Worku Mitiku, Muluken Azage Yenesew, Getasew Mulat Bantie, Amare Alamirew Aynie.

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
