## [Decision Letter · Decision Letter 0]

28 Jul 2020

PONE-D-20-15399

MAGNITUDE OF RISKY SEXUAL PRACTICE AND ASSOCIATED FACTORS AMONG BIG CONSTRUCTION SITE DAILY LABORERS IN BAHIR DAR CITY, AMHARA REGION, ETHIOPIA

PLOS ONE

Dear Dr. AYNIE,

Thank you for submitting your manuscript to PLOS ONE. After careful consideration, we feel that it has merit but does not fully meet PLOS ONE’s publication criteria as it currently stands. Therefore, we invite you to submit a revised version of the manuscript that addresses the points raised during the review process.

The reviewer has raised questions about the structure of the paper. I think that the issue of sexual behaviours vs practices needs at least some comment.

We look forward to receiving your revised manuscript.

Kind regards,

Andrew R. Dalby, PhD

Academic Editor

PLOS ONE

Journal Requirements:

2. Please address the following:

- Please include additional information regarding the survey or questionnaire used in the study and ensure that you have provided sufficient details that others could replicate the analyses. For instance, if you developed a questionnaire as part of this study and it is not under a copyright more restrictive than CC-BY, please include a copy, in both the original language and English, as Supporting Information. In addition, please provide further details of the content and development of this tool.

- Please ensure you have thoroughly discussed all potential limitations of this study within the Discussion section, including the impact of confounding factors and biases introduced.

- Please refrain from stating p values as 0.000, either report the exact value or employ the format p<0.001.

- Please provide additional details regarding participant consent. In the ethics statement in the Methods and online submission information, please ensure that you have specified how verbal consent was documented and witnessed.

3. Your ethics statement must appear in the Methods section of your manuscript. If your ethics statement is written in any section besides the Methods, please move it to the Methods section and delete it from any other section. Please also ensure that your ethics statement is included in your manuscript, as the ethics section of your online submission will not be published alongside your manuscript.

Reviewers' comments:

Reviewer's Responses to Questions

**Comments to the Author**

1. Is the manuscript technically sound, and do the data support the conclusions?

Reviewer #1: No

2. Has the statistical analysis been performed appropriately and rigorously? 

Reviewer #1: Yes

3. Have the authors made all data underlying the findings in their manuscript fully available?

Reviewer #1: Yes

4. Is the manuscript presented in an intelligible fashion and written in standard English?

Reviewer #1: No

5. Review Comments to the Author

Reviewer #1: 1. The introduction section lacks coherence. For instance, the fourth paragraph should be before the third one.

2. You have mentioned that most studies in Ethiopia and abroad were mainly focusing on risky sexual behaviors; but risky sexual practice was not well studied. Can you explain the difference between risky sexual behavior and risky sexual practice in your manuscript?

3. The justification for conducting this study is not clear. Please can you write something on what has been done so far by others in your country and beyond on similar or related topic. You only mentioned “The prevalence and its determinants of risky sexual behavior was documented before seven years by other investigators, but current risky sexual practice and its determining factors have not yet been explored.” To what extent are you confident to say current risky sexual practices have not been explored? Even if you are confident enough you must correct your statement by putting some uncertainty. Moreover, support what you wrote with references.

4. Under methodology you have used proportion of risky sexual behavior as 44.9 %. Does it is proportion of risky sexual behavior among daily laborers? If not it should be changed to prevalence of RSB among daily laborers in your study area or other place. Otherwise you should take 50%.

5. What was the rationale behind taking 20% of the big construction site? Please include under sampling technique and support with reference. In addition, you are expected to put the rationale for taking any variables with p-value of <0.2 on bivariate regression for multivariate analysis.

6. Can you include inclusion and exclusion criteria for the study?

7. You have operationalized several terminologies (e.g. big construction, risky sexual behavior etc). Have you defined these terminologies by yourself or you adopted definition given by someone or institution. If you adopted it, please write the references.

8. Do you have other justification for disparities with previous studies in Ethiopia regarding prevalence of risky sexual practices other than the time of study and sample size? Please justify the disparities in scientific way. Hopefully there more reasons for the difference.

9. With regard to factors associated with risky sexual practices instead of comparing you findings with previous similar studies you are expected to reason out why you got such findings.

10. Can you mention the new things your research brought to scientific community?

11. Generally this manuscript need language improvement

6. PLOS authors have the option to publish the peer review history of their article (what does this mean?). If published, this will include your full peer review and any attached files.

Reviewer #1: No

---

## [Author Response · Author response to Decision Letter 0]

6 Oct 2020

Dear editors and reviewers, 

Good Day to you all!

We authors have revised the manuscript as per your comments given.

And we thank you very much!

---

## [Editor Report · Decision Letter 1]

12 Oct 2020

MAGNITUDE OF RISKY SEXUAL PRACTICE AND ASSOCIATED FACTORS AMONG BIG CONSTRUCTION SITE DAILY LABORERS IN BAHIR DAR CITY, AMHARA REGION, ETHIOPIA

PONE-D-20-15399R1

Dear Dr. AYNIE,

We’re pleased to inform you that your manuscript has been judged scientifically suitable for publication and will be formally accepted for publication once it meets all outstanding technical requirements.

Kind regards,

Andrew R. Dalby, PhD

Academic Editor

PLOS ONE
---

## [Editor Report · Acceptance letter]

20 Oct 2020

PONE-D-20-15399R1 

*Magnitude of Risky Sexual Practice and Associated Factors Among Big Construction Site Daily Laborers in Bahir Dar City, Amhara Region, Ethiopia*

Dear Dr. AYNIE:

I'm pleased to inform you that your manuscript has been deemed suitable for publication in PLOS ONE. Congratulations! Your manuscript is now with our production department. 

Kind regards, 

on behalf of

Dr. Andrew R. Dalby 

Academic Editor

PLOS ONE